# Ling-CL: Understanding NLP Models through Linguistic Curricula

**Mohamed Elgaar** and **Hadi Amiri**
University of Massachusetts Lowell
{melgaar,hadi}@cs.uml.edu

## Abstract

We employ a characterization of linguistic complexity from psycholinguistic and language acquisition research to develop data-driven curricula to understand the underlying linguistic knowledge that models learn to address NLP tasks. The novelty of our approach is in the development of linguistic curricula derived from data, existing knowledge about linguistic complexity, and model behavior during training. By analyzing several benchmark NLP datasets, our curriculum learning approaches identify sets of linguistic metrics (indices) that inform the challenges and reasoning required to address each task. Our work will inform future research in all NLP areas, allowing linguistic complexity to be considered early in the research and development process. In addition, our work prompts an examination of gold standards and fair evaluation in NLP.

## 1 Introduction

Linguists devised effective approaches to determine the linguistic complexity of text data (Wolfe-Quintero et al., 1998; Bulté and Housen, 2012; Housen et al., 2019). There is a spectrum of *linguistic complexity indices* for English, ranging from lexical diversity (Malvern et al., 2004; Yu, 2010) to word sophistication (O'Dell et al., 2000; Harley and King, 1989) to higher-level metrics such as readability, coherence, and information entropy (van der Sluis and van den Broek, 2010). These indices have not been fully leveraged in NLP.

We investigate the explicit incorporation of linguistic complexity of text data into the training process of NLP models, aiming to uncover the linguistic knowledge that models learn to address NLP tasks. Figure 1 shows data distribution and accuracy trend of Roberta-large (Liu et al., 2019) against the linguistic complexity index "verb variation" (ratio of distinct verbs). This analysis is conducted on ANLI (Nie et al., 2020) validation data, where balanced accuracy scores are computed

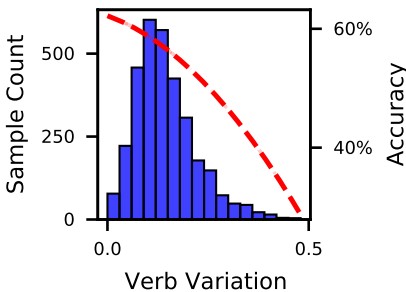

Figure 1: Data distribution and trend of model accuracy against the linguistic index *verb variation* computed on ANLI (Nie et al., 2020) validation data. Samples with greater verb variation are more complex and also harder for the model to classify. Such linguistic indices can inform difficulty estimation and linguistic curriculum development for NLP tasks.

for individual bins separately. The accuracy trend indicates that verb variation can describe the difficulty of ANLI samples to the model. In addition, the data distribution illustrates potential *linguistic disparity* in ANLI; see §3.4

To reveal the linguistic knowledge NLP models learn during their training, we will employ known linguistic complexity indices to build *multiview linguistic curricula* for NLP tasks. A curriculum is a training paradigm that schedules data samples in a meaningful order for iterative training, e.g., by starting with easier samples and gradually introducing more difficult ones (Bengio et al., 2009). Effective curricula improve learning in humans (Tabibian et al., 2019; Nishimura, 2018) and machines (Bengio et al., 2009; Kumar et al., 2010; Zhou et al., 2020; Castells et al., 2020). Curriculum learning has been found effective in many NLP tasks (Settles and Meeder, 2016; Amiri et al., 2017; Platanios et al., 2019; Zhang et al., 2019; Amiri, 2019; Xu et al., 2020; Lalor and Yu, 2020; Jafarpour et al., 2021; Kreutzer et al., 2021; Agrawal and Carpuat, 2022; Maharana and Bansal, 2022). A multiview curriculum is a curriculum able to integrate multiple difficulty scores simultaneously and leverage their collective value (Vakil and Amiri, 2023).

We assume there exists a subset of linguistic complexity indices that are most influential to learning an NLP task by a particular model. To identify these indices for each model and NLP task, we derive a weight factor $\rho_i \in [-1, 1]$ for each linguistic index that quantifies how well the index estimates the *true* difficulty of data samples to the model, determined by model instantaneous loss against *validation* data. By learning these weight factors, we obtain precise estimations that shed light on the core linguistic complexity indices that each model needs at different stages of its training to learn an NLP task. In addition, these estimates can be readily used for linguistic curriculum development, e.g., by training models with linguistically easy samples (with respect to the model) and gradually introducing linguistically challenging samples.

To achieve the above goals, we should address two gaps in the existing literature: First, existing curricula are often limited to a single criterion of difficulty and are not applicable to multiview settings. This is while difficulty is a condition that can be realized from multiple perspectives, can vary across a continuum for different models, and can dynamically change as the model improves. Second, existing approaches quantify the difficulty of data based on instantaneous *training* loss. However, training loss provides noisy estimates of sample difficulty due to data *memorization* (Zhang et al., 2017; Arpit et al., 2017) in neural models. We will address both issues as part of this research.

The contributions of this paper are:

- incorporating human-verified linguistic complexity information to establish an effective scoring function for assessing the difficulty of text data with respect to NLP models,
- deriving linguistic curricula for NLP models based on linguistic complexity of data and model behavior during training, and
- identifying the core sets of linguistic complexity indices that most contribute to learning NLP tasks by models.

We evaluate our approach on several NLP tasks that require significant linguistic knowledge and reasoning to be addressed. Experimental results show that our approach can uncover latent linguistic knowledge that is most important for addressing NLP tasks. In addition, our approach obtains consistent performance gain over competing models. Source code and data is available at https://github.com/CLU-UML/Ling-CL.

---

**Algorithm 1** Correlation Method

**Require:** $\mathcal{D}_{train}$, $\mathcal{D}_{val}$, Model $\Theta$, Optimizer $g$, Loss function $f$, Curriculum $C$
1: step $\leftarrow 0$
2: $\rho \leftarrow$ random initialization
3: **while** step $<$ total_steps **do**
4:     training_batch $\leftarrow$ SampleBatch(step, $\mathcal{D}_{train}$)
5:     loss $\leftarrow$ ComputeLoss (training_batch, $\Theta$)
6:     ling $\leftarrow$ GetLinguisticFeatures(training_batch)
7:     difficulty $\leftarrow$ CalculateDifficulty(ling, $\rho$)
8:     confidence $\leftarrow$ DetermineConfidence(step, difficulty)
9:     weighted_loss $\leftarrow$ loss $\otimes$ confidence
10:     $\Theta \leftarrow$ UpdateModel (weighted_loss, $\Theta$)
11:     **if** step % validation_step = 0 **then**
12:         $l \leftarrow$ ComputeLoss ($\mathcal{D}_{val}$, $\Theta$)
13:         ling $\leftarrow$ GetLinguisticFeatures($\mathcal{D}_{val}$)
14:         **for** $\rho_i \in \rho$ **do**
15:             $\rho_i \leftarrow$ pearsonr($l$, ling[:, $i$])
16:         **end for**
17:     **end if**
18:     step $\leftarrow$ step $+ 1$
19: **end while**

---

## 2 Multiview Linguistic Curricula

We present a framework for multiview curriculum learning using linguistic complexity indices. Our framework estimates the importance of various linguistic complexity indices, aggregates the resulting importance scores to determine the difficulty of samples for learning NLP tasks, and develops novel curricula for training models using complexity indices. The list of all indices used is in Appendix A.

### 2.1 Linguistic Index Importance Estimation

#### 2.1.1 Correlation Approach

Given linguistic indices $\{\mathbf{X}_j\}_{j=1}^k$ of $n$ data samples, where $k$ is the number of linguistic indices and $\mathbf{X}_j \in \mathbb{R}^n$, we start by standardizing the indices $\{\mathbf{Z}_j = \frac{X_j - \mu_j}{\sigma_j}\}_{j=1}^k$. We consider importance weight factors for indices $\{\rho_j\}_{j=1}^k$, which are randomly initialized at the start of training. At every validation step, the weights are estimated using the *validation* dataset by computing the Pearson's correlation coefficient between loss and linguistic indices of the validation samples $\rho_j = r(\mathbf{l}, \mathbf{Z}_j)$ where $r$ is the correlation function and $\mathbf{l} \in \mathbb{R}^n$ is the loss of validation samples. The correlation factors are updated periodically. It is important to use validation loss as opposed to training loss because the instantaneous loss of *seen* data might be affected by memorization in neural networks (Zhang et al., 2017; Arpit et al., 2017; Wang et al., 2020). This is while *unseen* data points more accurately represent the difficulty of samples for a model. Algorithm 1 presents the correlation approach.

### 2.1.2 Optimization Approach

Let $\mathbf{Z} \in \mathbb{R}^{n \times k}$ be the matrix of $k$ linguistic indices computed for $n$ validation samples and $\mathbf{l} \in \mathbb{R}^n$ indicate the corresponding loss vector of validation samples. We find the optimal weights for linguistic indices to best approximate validation loss:

$$\boldsymbol{\rho}^* = \arg\min_{\boldsymbol{\rho}} \quad \|\mathbf{l} - \mathbf{Z}\boldsymbol{\rho}\|_2^2 + \lambda_\rho \|\boldsymbol{\rho}\|_1, \quad (1)$$

where $\lambda_\rho \in \mathbb{R}$ and $\boldsymbol{\rho}^* \in \mathbb{R}^k$ is jointly optimized over all indices. The index that best correlates with loss can be obtained as follows:

$$i^* = \arg\min_i \|\mathbf{l} - \mathbf{Z}_{*i}\boldsymbol{\rho}_i\|_2^2, \quad (2)$$

where $\mathbf{Z}_{*i}$ denotes the $i^{th}$ column of $\mathbf{Z}$. Algorithm 2 presents this approach.

We note that the main distinction between the two correlation and optimization approaches lies in their scope: the correlation approach operates at the index level, whereas the optimization approach uses the entire set of indices.

### 2.1.3 Scoring Linguistic Complexity

We propose two methods for aggregating linguistic indices $\{X_j\}^k$ and their corresponding importance factors $\{\rho_j\}^k$ into a linguistic complexity score. The first method selects the linguistic index with the maximum importance score at each timestep:

$$S_i = Z_{i\hat{j}}, \quad \hat{j} = \arg\max_j \rho_j, \quad (3)$$

which provides insights into the specific indices that determine the complexity to the model.

The second method computes a weighted average of linguistic indices, which serves as a difficulty score. This is achieved as follows:

$$S_i = \frac{\sum_j \rho_j \mathbf{Z}_{ij}}{\sqrt{\sum_j \rho_j^2}}, \quad (4)$$

where $S_i \in \mathbb{R}$, $(\mu_{S_i}, \sigma_{S_i}) = (0, 1)$, is an aggregate of linguistic complexity indices for the input text. If an index $\mathbf{Z}_j$ is negatively correlated with loss, $\rho_j$ will be negative, and $\rho_j \mathbf{Z}_j$ will be positively correlated with loss. Therefore, $S_i$ is an aggregate complexity that is positively correlated with loss. And using weighted average results in the indices that are most highly correlated with loss to have the highest contribution to $S_i$.

---

**Algorithm 2** Optimization Method

**Require:** $\mathcal{D}_{train}, \mathcal{D}_{val}$, Model $\Theta$, Optimizer $g$, Optimizer $h$, Loss function $f$, [Optional] Curriculum $C$
1: step $\leftarrow 0$
2: $\rho \leftarrow$ random initialization
3: **while** step $<$ total_steps **do**
4:      training_batch $\leftarrow$ SampleBatch(step, $\mathcal{D}_{train}$)
5:      loss $\leftarrow$ ComputeLoss (training_batch, $\Theta$)
6:      ling $\leftarrow$ GetLinguisticFeatures(training_batch)
7:      difficulty $\leftarrow$ CalculateDifficulty(ling, $\rho$)
8:      confidence $\leftarrow$ DetermineConfidence(step, difficulty)
9:      weighted_loss $\leftarrow$ loss $\otimes$ confidence
10:     $\Theta \leftarrow$ UpdateModel (weighted_loss, $\Theta$)
11:     **if** step % validation_step = 0 **then**
12:        $l \leftarrow$ ComputeLoss $(\mathcal{D}_{val}, \Theta)$
13:        ling $\leftarrow$ GetLinguisticFeatures($\mathcal{D}_{val}$)
14:        $\rho \leftarrow \arg\min_\rho \|\text{ling} \cdot \rho - l\|_2^2 + \lambda_\rho \|\rho\|_1$
15:     **end if**
16:     step $\leftarrow$ step $+ 1$
17: **end while**

---

### 2.2 Linguistic Curriculum

We evaluate the quality of weighted linguistic indices as a difficulty score and introduce three new curricula based on a moving logistic (Richards, 1959) and Gaussian functions, see Figure 2.

### 2.2.1 Time-varying Sigmoid

We develop a time-varying sigmoid function to produce weights (Eq. 3). The sigmoid function assigns a low weight to samples with small difficulty scores and a high weight to larger difficulty scores. Weights are used to emphasize or de-emphasize the loss of different samples. For this purpose, we use the training progress $t \in [0, 1]$ as a shift parameter, to move the sigmoid function to the left throughout training, so that samples with a small difficulty score are assigned a higher weight in the later stages of training. By the end of the training, all samples are assigned a weight close to 1. Additionally, we add a scale parameter $\beta \in [1, \infty)$ that controls the growth rate of weight (upper bounded by 1) for all samples.

$$w(S_i, t; \beta) = \frac{1}{1 + \exp(-S_i - t \cdot \beta)}. \quad (5)$$

The sigmoid function saturates as the absolute value of its input increases. To account for this issue, our input aggregated linguistic index follows the standard scale, mean of zero, and variance of one, in (4) and (3).

### 2.2.2 Moving Negative-sigmoid

The positive-sigmoid function assigns greater weights to samples with a large value for $S$ that

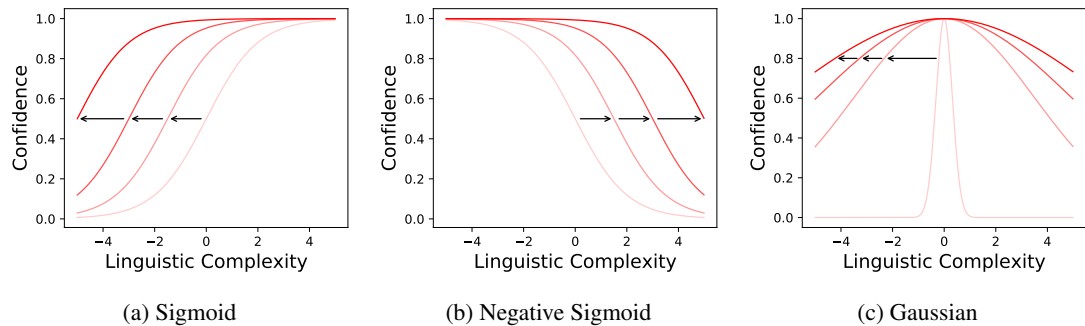

|          |                   |             |
|:--------:|:-----------------:|:-----------:|
| (a) Sigmoid | (b) Negative Sigmoid | (c) Gaussian |

Figure 2: At the beginning of training, the sigmoid function with the lowest opacity is used. It is the right-most curve in (a), the left-most curve in (b), and the middle curve in (c). Then, as training progresses, the function is shifted using the parameter $t$ in (5) and (7), causing samples with a higher complexity to be assigned higher confidence if (a) is used, samples with a lower complexity to be assigned higher confidence if (b) is used, and samples with medium complexity to be assigned higher confidence if (c) is used.

are linguistically more complex. In order to establish a curriculum that starts with easy samples and gradually proceeds to harder ones, we use a negative sigmoid function:

$$w(S_i, t; \beta) = \frac{1}{1 + \exp(S_i - t \cdot \beta)}. \quad (6)$$

Figure 2 illustrates the process of time-varying positive and negative sigmoid functions. Over the course of training, larger intervals of linguistic complexity are assigned full confidence, until the end of training when almost all samples have a confidence of one and are fully used in training.

### 2.2.3 Time-varying Gaussian Function

We hypothesize that training samples that are not too hard and not too easy are the most useful in training, and should receive the most focus. In fact, samples that are too easy or hard may contain artifacts that are harmful to training, may contain noise, and may not be generalizable to the target task. Therefore, we use the Gaussian function to prioritize learning from medium-level samples. The function starts with a variance of 1, and scales up during the course of training so that the easier and harder samples, having lower and higher linguistic complexity values, respectively, are assigned increasing weights and are learned by the end of training. We propose the following function:

$$w(S_i, t; \gamma) = \exp\left(\frac{-S_i^2}{2(1 + t \cdot \gamma)}\right), \quad (7)$$

where $\gamma$ is the rate of growth of variance and $t$ is the training progress, see Figure 2.

### 2.2.4 Weighting-based Curriculum

We define a curriculum by weighting sample losses according to their confidence. Samples that are most useful for training receive higher weights, and those that are redundant or noisy receive smaller weights. Weighting the losses effectively causes the gradient update direction to be dominated by the samples that the curriculum thinks are most useful. Weights $w$ are computed using either Equation 5, 6 or 7:

$$\mathcal{L} = \frac{1}{\sum_i w(S_i, t; \beta)} \sum_i w(S_i, t; \beta) \cdot \ell_i, \quad (8)$$

where $\ell_i$ is the loss of sample $i$, $t$ the current training progress, and $\mathcal{L}$ is the average weighted loss.

### 2.3 Reducing Redundancy in Indices

We have curated a list of 241 linguistic complexity indices. In the case of a text pair input (e.g. NLI), we concatenate the indices of the two text inputs, for a total of 482. Our initial data analysis reveals significant correlation among these indices in their estimation of linguistic complexity. To optimize computation, avoid redundancy, and ensure no single correlated index skews the complexity aggregation approach 2.1.3, we propose two methods to select a diverse and distinct set of indices for our study. We consider the choice of using full indices or filtering them as a hyper-parameter.

In the first approach, for each linguistic index, we split the dataset into $m$ partitions based on the index values [1] (similar to Figure 1). Next, using a trained No-CL (§3.3) model, we compute the accuracy for each partition. Then, we find the first-order

---

[1]We use numpy.histogram_bin_edges.

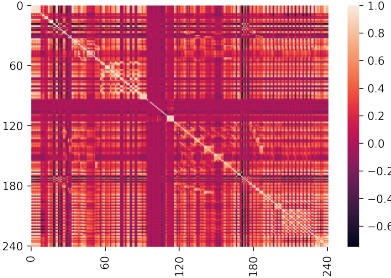

(a) Pair-wise correlation between indices

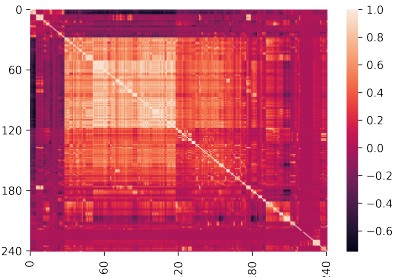

(b) Clustered correlation matrix.

Figure 3: Eliminating redundancy in linguistic indices. (a) shows the Pearson's correlation coefficient between each pair of linguistic indices. (b) is created by re-ordering the rows and columns of (a), such that mutually correlated indices are clustered into blocks using hierarchical clustering (Kumar et al., 2000). Best seen in color; lighter areas indicate greater correlations among index pairs or groups.

accuracy trend across these partitions. Linguistic indices with a pronounced slope describe great variance in the data and are considered for our study; we select the top 30% of indices, reducing their count from 482 to 144 for text pair inputs.

In the second approach, we compute pair-wise correlations between all indices. Then, we group highly correlated indices, as shown in Figure 3. From each cluster, we select a representative index, aiming to prevent correlated indices from dominating the aggregation approach and to eliminate redundancy. This method narrows our focus to the following 16 key indices: 1) type-token ratio (TTR), 2) semantic richness, 3) ratio of verbs to tokens, 4) mean TTR of all $k$ word segments, 5) Total number of verbs, 6) number of unique words, 7) adverbs per sentence, 8) number of unique words in the first $k$ tokens, 9) ratio of nouns to verbs, 10) semantic noise, 11) lexical sophistication, 12) verb sophistication, 13) clauses per sentence, 14) average SubtlexUS CDlow value per token, 15) adjective variation, 16) ratio of unique verbs. Please refer to Appendix A for definitions and references to indices.

# 3 Experiments

## 3.1 Datasets

We evaluate NLP models in learning the tasks of the following datasets:

- **SNLI**: Stanford Natural Language Inference (Bowman et al., 2015). The task is to classify a pair of sentences by the relation between them as one of entailment, neutral, or contradiction.

- **CoLA**: Corpus of Linguistic Acceptability (Warstadt et al., 2019). It is a task of classifying sentences as grammatical vs. ungrammatical.

- **ANLI**: Adverserial Natural Language Inference (Nie et al., 2020). This NLI dataset was created with a model in the loop, by only adding samples to the dataset that fool the model. We train only on the ANLI training set of 162k samples.

- **SST-2**: Stanford Sentiment Treebank (Socher et al., 2013). The task is to predict the sentiment of a given sentence as positive or negative.

- **RTE**: Recognizing Textual Entailment (Wang et al., 2018). The task is to determine if a given sentence is entailed by another given sentence.

- **AN-Pairs**: Adjective-noun pairs from the Cambridge ESOL First Certificate in English (FCE) exams (Kochmar and Briscoe, 2014). The task is to detect if an adjective-noun pair, including pairs that are typically confusing to language learners, is used correctly in the context of a sentence.

- **GED**: Grammatical Error Detection (Yannakoudakis et al., 2011). The task is to identify grammar errors at word level in given sentences.

## 3.2 Difficulty Scoring Functions

The curriculum learning approaches in §2.2 use difficulty scores or compute confidence to quantify sample difficulty in order to rank sentences. We use as difficulty scores: aggregate linguistic complexity **Ling**, see Section 2.1.3, and **Loss** (Xu et al., 2020; Wu et al., 2021; Zhou et al., 2020). We take the loss from a proxy model (No-CL in §3.3) by recording all samples losses two times per epoch during training and computing the sample-wise average.

### 3.3 Baselines

We consider a no-curriculum baseline as well as several recent curriculum learning approaches.

- **No-CL**: no-curriculum uses standard random mini-batch sampling from the whole dataset without sample weighting.

- **Sampling** (Bengio et al., 2009) uses the easiest subset of the dataset at each stage of training. Instead of randomly sampling a mini-batch from the whole dataset, a custom data sampler is created that provides the subset consisting of the easiest $\alpha\%$ of data when training progress is at $\alpha\%$.

- **SL-CL & WR-CL** (Platanios et al., 2019) is a curriculum learning approach that defines a time-varying function of the model's competence (defined as the fraction of training data that the model uses at every step), and a difficulty score of the data. At each iteration, a minibatch is sampled from the subset of data with difficulty smaller than the model's competence—a pre-defined non-linear function. The model employs sentence length (SL-CL) and word rarity (WR-CL) as difficulty measures. Sampling is the same as **Competence**-based curriculum learning with a linear competence function.

- **SuperLoss** (Castells et al., 2020) uses instantaneous loss to compute task-agnostic sample confidence. It emphasizes easy samples and de-emphasizes hard samples based on the global average loss as the difficulty threshold.

- **Concat** (Lee et al., 2021) concatenates linguistic indices to language embeddings before classification. Lee et al. (2021) and Meng et al. (2020) reported low performance as a result of appending features to embeddings. However, our approach succeeds in utilizing concatenated features.

- **Data Selection** (Mohiuddin et al., 2022) is an online curriculum learning approach. It evaluates the training data at every epoch and uses loss as the difficulty score. It selects the middle 40% of samples according to difficulty.

We compare the above models against our approaches, **Ling-CL**, which aggregates linguistic indices using weighted average or max-index aggregation, and applies different curriculum strategies:

sigmoid, negative-sigmoid, and Gaussian weighting, as well as sampling an competence-based approaches, see §3.3. We test variants of our approach with the correlation method, optimization method, and indices filtering. We report results of the *max* aggregation (§2.1.3) approach as it performs better than the weighted average and is computationally cheaper. **Loss-CL** computes loss as a difficulty score by recording the losses of samples during training of No-CL. The loss during the early stages of training generated by an under-trained model is a good measure of the relative difficulty of both training and validation samples.

### 3.4 Evaluation Metrics

Linguistic disparity can be quantified by the extent of *asymmetry* in the probability distribution of the linguistic complexity of samples in a dataset, e.g., see Figure 1 in §1. A natural solution to evaluate models is to group samples based on their linguistic complexity. Such grouping is crucial because if easy samples are overrepresented in a dataset, then models can result in unrealistically high performance on that dataset. Therefore, we propose to partition datasets based on a difficulty metric (linguistic index or loss) and compute balanced accuracy of different models on the resulting groups. This evaluation approach reveals great weaknesses in models, and benchmark datasets or tasks that seemed almost "solved" such as as the complex tasks of NLI.

### 3.5 Experimental Settings

We use the transformer model *roberta-base* (Liu et al., 2019) from (Wolf et al., 2020), and run each experiment with at least two random seeds and report the average performance. We use AdamW (Loshchilov and Hutter, 2018) optimizer with a learning rate of $1 \times 10^{-5}$, batch size of 16, and weight decay of $1 \times 10^{-2}$ for all models. The model checkpoint with the best validation accuracy is used for final evaluation. In NLI tasks with a pair of text inputs, the indices of both texts are used. For Ling-CL, we optimize the choice of index importance estimation method and aggregation method. For the baselines, we optimize the parameters of SuperLoss ($\lambda$ and moving average method), and the two parameters of SL-CL and WR-CL models for each dataset. For the data selection, we use a warm-up period of 20% of the total training iterations.

| | ANLI | COLA | RTE | SNLI | SST2 | AN-Pairs | GED | Average |
|---|---|---|---|---|---|---|---|---|
| **Ling-CL [NegSig]** | $\underline{59.3}$ ± 2.55 | $\underline{72.4}$ ± 0.40 | **79.1** ± 8.47 | 82.8 ± 8.35 | 92.2 ± 0.22 | 79.1 ± 1.55 | 75.3 ± 0.67 | 77.2 ± 3.17 |
| **Ling-CL [Gauss]** | **60.9** ± 1.41 | **73.0** ± 0.02 | 77.2 ± 8.08 | $\underline{83.5}$ ± 8.39 | $\underline{92.4}$ ± 0.27 | 82.9 ± 1.24 | $\underline{75.5}$ ± 0.41 | **77.9** ± 2.83 |
| **Ling-CL [Sig]** | 58.1 ± 0.17 | 64.6 ± 8.91 | $\underline{78.7}$ ± 8.87 | 83.0 ± 8.48 | 92.3 ± 0.01 | 82.3 ± 0.93 | **75.9** ± 0.10 | 76.4 ± 3.92 |
| **Loss-CL [NegSig]** | 59.0 ± 0.31 | 55.6 ± 0.64 | 68.1 ± 1.59 | 75.1 ± 0.05 | 91.6 ± 0.26 | 76.4 ± 5.70 | 75.1 ± 1.44 | 71.6 ± 1.43 |
| **Loss-CL [Sig]** | 49.7 ± 9.58 | 56.6 ± 0.37 | 66.8 ± 0.29 | **83.6** ± 8.37 | 90.9 ± 0.42 | 81.4 ± 0.61 | 73.3 ± 0.29 | 71.8 ± 2.85 |
| **Loss-CL [Gauss]** | 49.4 ± 11.07 | 57.0 ± 1.29 | 67.2 ± 1.41 | 75.1 ± 0.52 | 91.8 ± 0.12 | 80.5 ± 2.08 | 74.5 ± 0.07 | 70.8 ± 2.37 |
| **Sampling** | 49.9 ± 10.00 | 64.6 ± 8.89 | 67.9 ± 0.03 | 83.2 ± 8.72 | 91.5 ± 0.07 | 82.6 ± 3.93 | 73.8 ± 1.23 | 73.4 ± 4.7 |
| **Competence** | 50.1 ± 11.27 | 63.4 ± 9.08 | 68.8 ± 0.64 | 74.7 ± 0.06 | 91.6 ± 0.03 | **84.0** ± 1.14 | 74.1 ± 0.39 | 72.4 ± 3.23 |
| **SL-CL** | 50.3 ± 10.05 | 55.8 ± 0.06 | 67.7 ± 1.27 | 82.6 ± 8.35 | **93.1** ± 0.00 | 81.6 ± 0.72 | 75.2 ± 0.26 | 72.3 ± 2.96 |
| **WR-CL** | 50.9 ± 9.80 | 56.1 ± 0.53 | 68.4 ± 0.73 | 74.5 ± 0.16 | 91.5 ± 0.16 | 80.1 ± 0.81 | 75.2 ± 0.17 | 71.0 ± 1.77 |
| **SuperLoss** | 39.5 ± 0.14 | 56.9 ± 0.69 | 69.6 ± 0.50 | 75.2 ± 0.14 | 91.7 ± 0.26 | 77.8 ± 1.89 | 74.2 ± 0.15 | 69.3 ± 0.54 |
| **Concat** | 51.3 ± 9.83 | 64.3 ± 8.03 | 71.4 ± 0.51 | 75.2 ± 0.24 | 91.9 ± 0.14 | 81.8 ± 1.66 | 73.8 ± 0.91 | 72.8 ± 3.05 |
| **Data Selection** | 46.8 ± 6.12 | 55.1 ± 1.71 | 66.6 ± 1.49 | 74.4 ± 0.49 | 91.5 ± 0.30 | 79.6 ± 1.03 | $\underline{75.5}$ ± 0.52 | 69.9 ± 1.67 |
| **No-CL** | 51.7 ± 8.21 | 57.0 ± 0.22 | 70.0 ± 0.45 | 83.3 ± 8.42 | 83.7 ± 8.22 | 82.1 ± 0.51 | 74.0 ± 0.14 | 71.7 ± 3.74 |

Table 1: Balanced accuracy by linguistic index (Word rarity). Accuracy is the metric for all datasets except CoLA and GED, CoLA uses Matthew's correlation and GED uses $F_{\beta=0.5}$ score. Ling-CL uses aggregate linguistic complexity as a difficulty score we create, and Loss-CL uses the average loss of a sample throughout a full training.

## 3.6 Enhanced Linguistic Performance

Tables 1 show the performance of different models when test samples are grouped based on word rarity. The results show that the performance of the baseline models severely drops compared to standard training (No-CL). This is while our Ling-CL approach results in 4.5 absolute points improvement in accuracy over the best-performing baseline averaged across tasks, owing to its effective use of linguistic indices. Appendix D shows the overall results on the entire test sets, and results when test samples are grouped based on their loss; we use loss because it is a widely-used measure of difficulty in curriculum learning. These groupings allow for a detailed examination of the model's performance across samples with varying difficulty, providing insights into the strengths and weaknesses of models. For example, the performance on SNLI varies from 89.8 to 90.6. However, when word rarity is used to group data based on difficulty, the performance range significantly drops from 74.4 to 83.6, indicating the importance of the proposed measure of evaluation. We observe that such grouping does not considerably change the performance on ANLI, which indicates the high quality of the dataset. In addition, it increases model performance on AN-Pair and GED, which indicates a greater prevalence of harder examples in these datasets.

On average, the optimization approach outperforms correlation by 1.6% ±1.9% accuracy in our experiments. Also notably, on average, the argmax index aggregation outperforms the weighted average by 1.9% ±1.9%, and the filtered indices outperform the full list of indices by 1.4% ±1.1%.

## 3.7 Learning Dynamics for NLP Tasks

**Identification of Key Linguistic Indices** We analyze the linguistic indices that most contribute to learning NLP tasks. For this purpose, we use the evaluation approach described in §3.4 for computing balanced accuracy according to linguistic indices. Table 2 shows the top three important linguistic indices for each dataset as identified by our optimization algorithm using the Gaussian curriculum. Importance is measured by the average $\rho$ value. Early, middle, and late divide the training progress into three equal thirds. The top index in the early stage is the index with the highest average $\rho$ during the first 33.3% of training. The top indices are those that most accurately estimate the true difficulty of samples, as they should highly correlate with validation loss.

Table 2 shows that different indices are important for different tasks. This means that it is not possible to use a single set of linguistic indices as a general text difficulty score, important indices can be identified for each task, which can be achieved by our index importance estimation approach (§2.1) and evaluation metric (§3.4).

**Analysis of Linguistic Indices for Grammar Tasks** We consider the grammar tasks for analysis. For AN-Pairs (adjective-noun pair), during the early stage, the top indices are the number of tokens per sentence, age of acquisition (AoA) of words, and mean length of sentences. This is meaningful because longer sentences might introduce modifiers or sub-clauses that can create ambiguity or make it more challenging to discern the intended adjective-noun relationship accurately. Regarding AoA, words that are acquired later in life or belong

| | **Early** | **Middle** | **Late** |
|---|---|---|---|
| **AN-Pairs** | # Tokens per sentence | Lemmas age of acquisition | # Adverbs per sentence |
| | Lemmas age of acquisition | # Tokens per sentence | Corrected TTR |
| | Mean sentence length | # Adverbs per sentence | Nouns to adjective ratio |
| **GED** | Corrected noun variation | | |
| | # Tokens per sentence | # Nouns per sentence | # Tokens per sentence |
| | # Nouns per sentence | # Tokens per sentence | # Nouns per sentence |
| **RTE** | Ratio of Adverbs to Verbs (P) | | |
| | Ratio of Subordinating Conjunctions to Verbs (P) | | Adverb Variation (P) |
| | Verb sophistication (P) | | Adverbs per sentence (P) |
| **ANLI** | Lexical verb variation (P) | | Function words per sentence (H) |
| | Unique Entities (P) | | Log Tokens per log sentences |
| | Unique Entities per token (P) | | |
| **SST-2** | # Complex nominals | | |
| | Noun variation | | |
| | Ratio of nouns to verbs | | Verb variation |
| **CoLA** | # Function words | | # Coordinating Conjunctions |
| | Number of T-units | | |
| | T-units per sentence | | |
| **SNLI** | Lemmas age of acquisition (P) | | |
| | Linsear Write Formula Score (P) | | |
| | Gunning Fog Count Score (P) | | |

Table 2: Top three important linguistic indices at each stage of learning. For datasets with a premise (P) and hypothesis (H), they are indicated in parentheses.

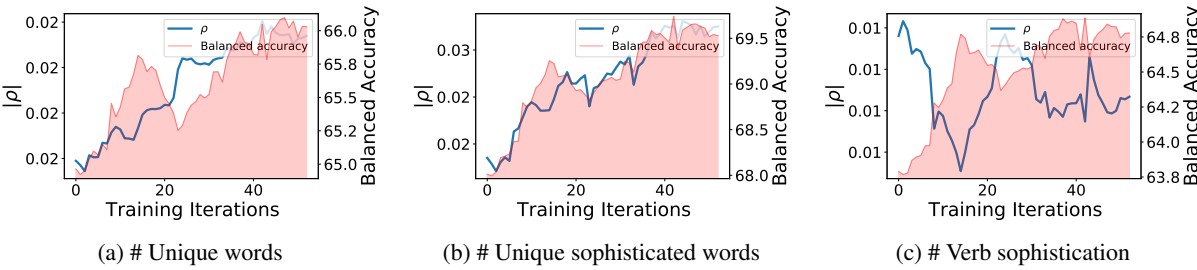

(a) # Unique words    (b) # Unique sophisticated words    (c) # Verb sophistication

Figure 4: The progression of the estimated importance factors $\rho$, and balanced accuracy for groups of linguistic indices.

to more specialized domains might pose challenges in accurately judging the correct usage of adjective-noun pairs because of their varying degrees of familiarity and potential difficulty associated with specific vocabulary choices.

During the middle stage the AoA increases in importance and remains challenging to the model, the number of adverbs per sentence increases in rank and joins the top three indices. In the context of adjective-noun pairs, the presence of multiple adverbs in a sentence can potentially affect the interpretation and intensity of the adjective's meaning. This is because adverbs often modify verbs, adjectives, or other adverbs in sentences. In addition, depending on the specific adverbs used, they may enhance, weaken, or alter the intended relationship between the adjective and the noun. Moreover, the presence of several adverbs can simply introduce potential challenges in identifying and correctly interpreting the relationship between adjectives and nouns due to increasing syntactic complexity.

In the third stage, the number of adverbs per sentence becomes the top important index, while AoA and the number of tokens per sentence drop out of the top three. In the early stage, AoA and the number of tokens has $\rho$ values of 0.168 and 0.164, respectively. In the late stage, they drop to 0.11 and 0.13, while the number of adverbs per sentence is 0.138 early, and increases to 0.181 in the late stage. We see that indices may become dominant not only by increasing their $\rho$ value but also by waiting for other indices to drop down when they have been learned by the model. Therefore, Ling-CL can determine the order to learn linguistic indices, and then learn them sequentially.

Regarding GED, noun variation is the dominant index throughout the training process. Such variation is important because it affects syntactic agreement, subject-verb agreement, modifier placement, and determiner selection. These factors affect gram-

matical consistency and coherence within the sentence structure, leading to the importance of noun variation throughout the training process.

**Dominant Indices for CoLA Task** Regarding CoLA, the number of function words and coordinating conjunctions indices are the dominant indices at the early stage, and middle and late stages of training respectively. These words are crucial in establishing the syntactic structure of a sentence. They directly contribute to agreement and references, coherence, and adherence to grammar rules. We note that T-units (independent/main clauses clauses with their associated subordinate clauses) are higher-order linguistic constituents that provide information about the dependency relations between sub-constituents, and the overall coherence of sentences. Indices related to T-units are among the top three crucial indices.

**Trends and Relationships between $\rho$ and Balanced Accuracy** We use the GED dataset (§3.1) to analyze the trends of $\rho$ throughout training, and the relation between $\rho$ and balanced accuracy. Figure 4 shows the progression of $\rho$ with the progression of balanced accuracy for selected linguistic indices. This figure is produced using No-CL. We observe across several indices that $\rho$ is high when balanced accuracy is low, indicating that the index is challenging to the model and therefore used for learning with a high $\rho$, and decreases as the index is learned. However, Figure 4a shows that it is not necessary that when balanced accuracy increases $\rho$ would decrease. In this case, it means that the model is performing relatively well on the index, but the index remains predictive of loss. So, although the average performance increased, the variation in performance among different values of the index remains high. We find that numerous indices follow the same of trend of $\rho$. In Appendix B, we propose a method for clustering $\rho$ to effectively uncover patterns and similarities in the learning of different indices. However, further analysis of the dynamics of $\rho$ is the subject of our future work.

In addition, we find that the rank of top indices is almost constant throughout the training. This quality may be useful in creating an approach that gathers the indices rankings early on and utilizes them for training. Appendix E lists influential indices by their change in $\rho$ across stages of training. We note that the "number of input sentences" index is the least important metric because the index is al-

most constant across samples—75% of the samples consist of a single sentence in the datasets.

## 4 Conclusion and Future Work

We propose a new approach to linguistic curriculum learning. Our approach estimates the importance of multiple linguistic indices and aggregates them, provides effective difficulty estimates through correlation and optimization methods, and introduces novel curricula for using difficulty estimates, to uncover the underlying linguistic knowledge that NLP models learn during training. Furthermore, we present a method for a more accurate and fair evaluation of computational models for NLP tasks according to linguistic indices. Furthermore, the estimated importance factors present insights about each dataset and NLP task, the linguistic challenges contained within each task, and the factors that most contribute to model performance on the task. Further analysis of such learning dynamics for each NLP task will shed light on the linguistic capabilities of computational models at different stages of their training.

Our framework and the corresponding tools serve as a guide for assessing linguistic complexity for various NLP tasks and uncover the learning dynamics of the corresponding NLP models during training. While we conducted our analysis on seven tasks and extracted insights on the key indices for each task, NLP researchers have the flexibility to either build on our results or apply our approach to other NLP tasks to extract relevant insights. Promising areas for future work include investigations on deriving optimal linguistic curriculum tailored for each NLP task; examining and enhancing linguistic capabilities of different computational models, particularly with respect to linguistically complex inputs; and developing challenge datasets that carry a fair distribution of linguistically complex examples for various NLP tasks. In addition, future work could study why specific indices are important, how they connect to the linguistic challenges of each task, and how different linguistic indices jointly contribute to learning a target task. We expect other aggregation functions, such as log-average, exponential-average, and probabilistic selection of maximum, to be effective approaches for difficulty estimation based on validation loss. Finally, other variations of the proposed Gaussian curriculum could be investigated for model improvement.

## 5 Limitations

Our work requires the availability of linguistic indices, which in turn requires expert knowledge. Such availability requirements may not be fulfilled in many languages. Nevertheless, some linguistic complexity indices are language independent, such as the commonly-used "word rarity" measure, which facilitates extending our approach to other languages. Moreover, our approach relies on the effectiveness of specific linguistic complexity indices for target tasks and datasets employed for evaluation; different linguistic complexity indices may not capture all aspects of linguistic complexity and may yield different results for the same task or dataset. In addition, the incorporation of linguistic complexity indices and the generation of data-driven curricula can introduce additional computational overhead during the training process. Finally, our approach does not provide insights into the the interactions between linguistic indices during training.

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

# A List of indices

| | |
|---|---|
| # Unique words | # Unique sophisticated words |
| # Unique lexical words | # Unique sophisticated lexical words |
| # Total words | # Total sophisticated words |
| # Total lexical words | # Total sophisticated lexical words |
| Lexical density | Lexical sophistication (total) |
| Lexical sophistication (unique) | Verb sophistication |
| Verb sophistication (squared numerator) | Verb sophistication (sqrt denominator) |
| Type-token ratio (TTR) | Mean TTR of all k word segments |
| Corrected TTR (sqrt(2N) denominator) | Root TTR (sqrt(N) denominator) |
| Log TTR | Uber index |
| Noun variation | Adjective variation |
| Adverb variation | (Ajd + Adv) variation |
| D Measure | Ratio of unique verbs |
| Verb variation with squared numerator | Verb variation with (sqrt(2N)) denominator |
| Verb variation over all lexical words | Unique words in first k tokens |
| Unique words in random k tokens | Unique words in random sequence of k tokens |

Table 3: Lexical indices

| | |
|---|---|
| # Words | # Sentences |
| # Verb phrases | # Clauses |
| # T-units | # Dependent clauses |
| # Complex T-units | # Coordinate phrases |
| # Complex nominals | Mean length of sentence |
| Mean length of T-unit | Mean unit of clause |
| Clauses per sentence | Verb phrases per T-unit |
| Clauses per T-unit | Dependent clause ratio |
| Dependent clause per T-unit | T-units per sentence |
| Complex T-unit ratio | Coordinate phrases per T-unit |
| Coordinate phrases per clause | Complex nominals per T-unit |
| Complex nominals per clause | |

Table 4: Syntactic indices

In our work we make use indices from Lu (2010), Lu (2012), and Lee et al. (2021). Table 3 lists the lexical indices (33 indices) and table 4 lists the syntactic indices (23 indices) that we use. For their full descriptions please refer to Lu (2010) and Lu (2012). In this section, we provide descriptions of a few relevant indices. Please refer to Lee et al. (2021) for the comprehensive list of lingfeat (185 indices) indices.

**TTR** is the ratio of unique words in the text. **D-measure** is a modification to TTR that is not biased by sample size. **Lexical words** are nouns, verbs, adjectives, and adverbs. **Sophisticated words** are the unconventional words. We consider words beyond the 2000 most frequent words in the American National Corpus as sophisticated. **Uber index** is a transformation of TTR. **SubtlexUS CDlow** is a word frequency measure, specifically, "document frequency" of words starting with a lower case letter.

# B  Clustering $\rho$

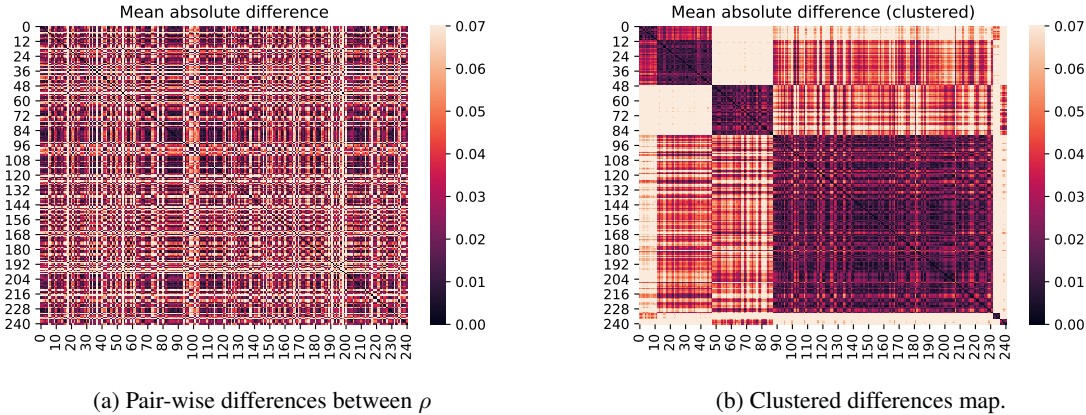

(a) Pair-wise differences between $\rho$

(b) Clustered differences map.

Figure 5: Figure (a) shows mean absolute difference between $\rho$ of each pair of linguistic indices, averaged over the whole training. Figure (b) is created by re-ordering the rows and columns of (a), such that groups of indices have minimal difference between them.

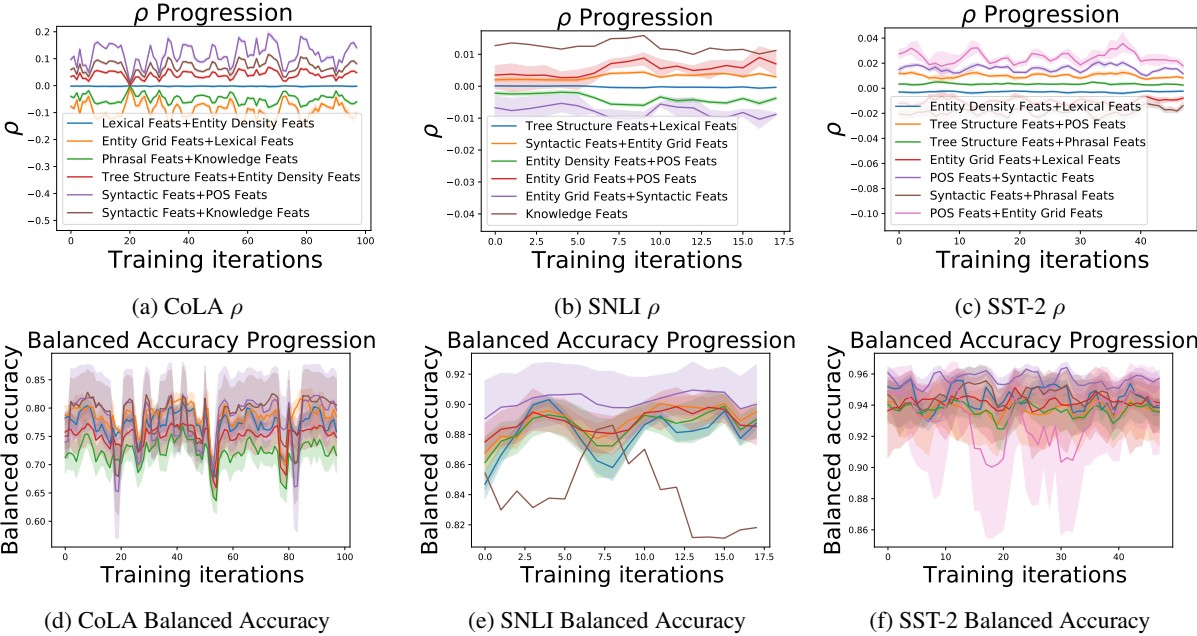

(a) CoLA $\rho$

(b) SNLI $\rho$

(c) SST-2 $\rho$

(d) CoLA Balanced Accuracy

(e) SNLI Balanced Accuracy

(f) SST-2 Balanced Accuracy

Figure 6: The progression of the estimated importance factors $\rho$, and balanced accuracy for groups of linguistic indices. Each pair of figures (a) and (d), (b) and (e), (c) and (f) share the legend. The solid line is the mean value of the group of lines, and the shaded area is the 95% confidence interval.

We observe that several indices follow the same patterns. Therefore, we devise a method to group indices that follow the same pattern of $\rho$. We compute the mean absolute difference between $\rho$ of each pair of linguistic indices. Then, we cluster the groups of indices that all have a minimum distance between them. Appendix B displays the effect of clustering. Note that the common trends among lines in Figures 6a, 6b, and 6c are because they are all governed by the trend of the validation loss (using both optimization and correlation approaches). Figures 6d, 6e, and 6f show the trend of balanced accuracy for the same groups. The grouped balanced accuracy has a very high variance. It shows that indices with similar $\rho$ do not have similar values of balanced accuracy. Moreover, it shows that indices with the highest $\rho$ do not necessarily have the highest mean balanced accuracy. Furthermore, indices that have $\rho = 0$ perform comparably to other indices, indicating that the model performs well according to such linguistic

indices, despite them not being correlated with loss.

Figure 5 illustrates the process of clustering together linguistic indices based on their matching $\rho$ curves. We cluster the indices using hierarchical clustering with *complete* linkage using the flat clustering method[2][3].

## C   Linguistic Complexity Indices

We consider linguistic complexity in terms of variability and sophistication in productive vocabulary and grammatical structures in textual content. We employ a characterization of such complexity based on existing findings in language acquisition research (Wolfe-Quintero et al., 1998; Lu, 2010, 2012). Specifically, we obtain 56 complexity measures from Lu (2010) and Lu (2012), including lexical and syntactic measures. Additionally, we use 185 linguistic features from the `lingfeat` library (Lee et al., 2021), including semantic, lexical, syntactic, discourse, and traditional features. In total, we use 241 indices. For inputs that consist of a pair of sentences, we concatenate the indices for a total of 482 indices.

### C.1   Lexical Complexity

In terms of lexical complexity, we consider three dimensions: lexical *density*, *sophistication*, and *variation* described below:

**Lexical density:**   is quantified by the ratio of the number of *open-class* words to the total number of words in a given text. Texts with higher lexical density are expected to be more complex as they contain larger amounts of information-carrying words.

**Lexical sophistication:**   measures the proportion of *sophisticated*—relatively unusual or advanced— words in the input text (O'Dell et al., 2000), e.g., words not in the top K (K= 5000) frequent words in the target dataset or language. Example indices include the ratio of the number of sophisticated lexical words (Linnarud, 1986; Hyltenstam, 1988), sophisticated word types (Wolfe-Quintero et al., 1998) and sophisticated verb types (Harley and King, 1989) in texts, which include several variations as reported in Appendix A, Table 3. We use the top K most frequent words of each dataset and consider different inflections of the same lemma as one type for computing lexical sophistication.

**Lexical Variation:**   refers to the diversity of vocabulary in a given text. Examples of such variations include type-token ratio (Templin, 1957) which is the ratio of the number of word types to the number of words in the text and several different variations of this metric (Malvern et al., 2004; McKee et al., 2000; McClure, 1991) including *D-measure* (Malvern et al., 2004), which determines lexical variation of an input text by finding the curve that best matches the actual curve of type-token ratio against tokens of the input.

### C.2   Syntactic Complexity

Syntactic complexity determines variability and sophistication with respect to grammatical structures. Simple sentences such as "*the mouse ate the cheese*" can be converted to their linguistically-complex counterparts, e.g., "*the mouse the cat the dog bit chased ate the cheese,*" which are still well-formed sentences but force readers to suspend their partial understanding of the entire sentence by encountering subordinate clauses that substantially increase the cognitive load of the sentences. We employ syntactic complexity measures that quantify the length of production units at the clausal, sentential, or T-unit levels; indices that reflect the amount of subordination, e.g., T-unit complexity ratio (clauses per T-unit) or dependent clause ratio (dependent clauses per clause); indices that quantify the amount of coordination, e.g., number of coordinate phrases per clause, T-unit or complex T-unit; as well as those that quantify the range of surface and particular syntactic and morphological structures (e.g., frequency and variety of tensed forms or extent of affixation) (Wolfe-Quintero et al., 1998; Ortega, 2003). See Appendix A, Table 3.

---

[2]`https://docs.scipy.org/doc/scipy/reference/generated/scipy.cluster.hierarchy.linkage.html`
[3]`https://docs.scipy.org/doc/scipy/reference/generated/scipy.cluster.hierarchy.fcluster.html`

# D   Full results

Tables 5 and 6 show the overall performance and performance balanced by loss. Our Ling-CL approach results in 1.3 absolute points improvement in accuracy over the best-performing baseline averaged across tasks, balanced by loss.

| | ANLI | COLA | RTE | SNLI | SST2 | AN-Pairs | GED | Average |
|---|---|---|---|---|---|---|---|---|
| **Ling-CL [NegSig]** | 49.6 ± 0.44 | 62.8 ± 0.19 | 81.0 ± 0.10 | 90.0 ± 0.10 | 95.0 ± 0.06 | 74.0 ± 1.04 | 71.6 ± 0.37 | 74.9 ± 0.33 |
| **Ling-CL [Gauss]** | 51.5 ± 0.39 | 64.5 ± 0.35 | 79.8 ± 0.21 | 90.4 ± 0.01 | **95.2** ± 0.00 | **75.0** ± 2.08 | 71.9 ± 0.20 | 75.5 ± 0.46 |
| **Ling-CL [Sig]** | 51.6 ± 0.17 | 65.1 ± 0.71 | 81.9 ± 0.21 | 90.2 ± 0.07 | 95.1 ± 0.11 | 74.5 ± 2.60 | 71.8 ± 0.05 | **75.7** ± 0.56 |
| **Loss-CL [NegSig]** | **52.7** ± 0.03 | 63.5 ± 0.66 | 80.5 ± 0.36 | 90.5 ± 0.12 | 95.0 ± 0.00 | 69.3 ± 1.54 | **72.2** ± 0.03 | 74.8 ± 0.39 |
| **Loss-CL [Gauss]** | 51.9 ± 0.06 | **65.7** ± 1.66 | **82.3** ± 1.08 | 90.3 ± 0.38 | 94.7 ± 0.11 | 72.9 ± 1.04 | 72.1 ± 0.08 | 75.7 ± 0.63 |
| **Loss-CL [Sig]** | 51.1 ± 1.20 | 65.0 ± 0.57 | 78.9 ± 1.26 | 90.4 ± 0.26 | 94.9 ± 0.06 | 74.3 ± 1.93 | 71.7 ± 0.15 | 75.2 ± 0.78 |
| **Sampling** | 49.3 ± 0.33 | 64.0 ± 1.02 | 78.4 ± 0.70 | 90.3 ± 0.14 | 94.6 ± 0.10 | 72.9 ± 4.17 | 71.1 ± 0.07 | 74.4 ± 0.93 |
| **Competence** | 50.7 ± 0.03 | 63.2 ± 0.63 | 77.8 ± 0.73 | 90.3 ± 0.17 | 95.0 ± 0.11 | 74.5 ± 0.52 | 71.2 ± 0.10 | 74.7 ± 0.33 |
| **SL-CL** | 51.1 ± 0.95 | 63.6 ± 0.42 | 80.3 ± 2.35 | 90.1 ± 0.11 | 94.9 ± 0.06 | 69.3 ± 1.56 | 71.9 ± 0.12 | 74.5 ± 0.80 |
| **WR-CL** | 51.9 ± 0.16 | 64.5 ± 1.00 | 80.0 ± 0.54 | 90.2 ± 0.18 | 94.4 ± 0.23 | 64.6 ± 4.17 | 71.7 ± 0.32 | 73.9 ± 0.94 |
| **Concat** | 52.2 ± 0.30 | 65.2 ± 0.33 | 81.8 ± 0.90 | **90.6** ± 0.00 | 94.6 ± 0.11 | 72.4 ± 2.60 | 71.7 ± 0.21 | 75.5 ± 0.64 |
| **SuperLoss** | 51.7 ± 0.21 | 64.3 ± 0.99 | 80.5 ± 1.04 | 90.5 ± 0.15 | 94.8 ± 0.13 | 67.7 ± 2.08 | 71.7 ± 0.10 | 74.5 ± 0.67 |
| **Data Selection** | 48.5 ± 0.94 | 59.4 ± 0.03 | 79.2 ± 0.90 | 89.8 ± 0.12 | 94.2 ± 0.17 | 72.4 ± 1.56 | 71.1 ± 0.13 | 73.5 ± 0.55 |
| **No-CL** | 51.4 ± 0.06 | 64.1 ± 0.42 | 81.4 ± 0.77 | 90.4 ± 0.17 | 94.9 ± 0.10 | 71.9 ± 2.08 | 72.0 ± 0.40 | 75.2 ± 0.57 |

Table 5: Overall performance of each approach. Unlike Tables 1 and 6, this table presents the standard un-balanced accuracy.

| | ANLI | COLA | RTE | SNLI | SST2 | AN-Pairs | GED | Average |
|---|---|---|---|---|---|---|---|---|
| **Ling-CL [NegSig]** | **23.2** ± 4.61 | 27.1 ± 0.66 | **45.4** ± 6.23 | 26.3 ± 0.44 | 25.7 ± 1.31 | 36.7 ± 4.98 | 73.9 ± 0.83 | 36.9 ± 2.72 |
| **Ling-CL [Gauss]** | 21.0 ± 1.89 | 26.7 ± 1.62 | 37.1 ± 0.52 | **42.9** ± 0.12 | 22.7 ± 2.91 | 34.7 ± 6.92 | 73.9 ± 0.36 | 37.0 ± 2.05 |
| **Ling-CL [Sig]** | 22.2 ± 0.51 | 26.4 ± 0.14 | 38.5 ± 0.27 | 35.2 ± 8.99 | 25.0 ± 0.58 | **38.4** ± 3.26 | 74.6 ± 0.11 | **37.2** ± 1.98 |
| **Loss-CL [NegSig]** | 20.9 ± 1.59 | 26.7 ± 0.42 | 37.7 ± 1.03 | 33.8 ± 8.37 | 23.8 ± 3.13 | 29.8 ± 3.76 | 73.4 ± 0.84 | 35.2 ± 2.73 |
| **Loss-CL [Sig]** | 19.5 ± 0.21 | 27.6 ± 0.51 | 36.3 ± 2.43 | 34.6 ± 7.58 | 24.2 ± 3.59 | 31.6 ± 1.95 | 72.6 ± 0.01 | 35.2 ± 2.33 |
| **Loss-CL [Gauss]** | 20.6 ± 0.99 | 26.3 ± 1.29 | 40.8 ± 3.89 | 26.2 ± 0.18 | 23.1 ± 1.63 | 31.7 ± 3.46 | 73.0 ± 0.38 | 34.5 ± 1.69 |
| **Sampling** | 19.0 ± 0.12 | 26.2 ± 0.36 | 32.9 ± 0.75 | 42.8 ± 0.50 | 24.1 ± 3.57 | 33.6 ± 1.09 | 72.8 ± 1.44 | 35.9 ± 1.12 |
| **Competence** | 21.5 ± 0.55 | 26.8 ± 1.84 | 36.9 ± 2.57 | 27.0 ± 0.34 | 27.1 ± 0.63 | 35.5 ± 2.01 | 73.4 ± 1.06 | 35.5 ± 1.29 |
| **SL-CL** | 21.3 ± 0.28 | 26.2 ± 0.00 | 31.6 ± 0.58 | 34.6 ± 8.35 | 26.7 ± 0.00 | 32.3 ± 2.59 | **74.8** ± 0.77 | 35.4 ± 1.8 |
| **WR-CL** | 19.3 ± 0.52 | 26.4 ± 0.69 | 35.7 ± 0.51 | 26.1 ± 0.03 | 24.2 ± 1.26 | 29.0 ± 0.73 | 73.5 ± 0.61 | 33.5 ± 0.62 |
| **SuperLoss** | 20.7 ± 1.77 | **29.0** ± 2.51 | 35.6 ± 0.99 | 26.0 ± 0.38 | 23.9 ± 0.83 | 25.9 ± 2.28 | 72.8 ± 0.03 | 33.4 ± 1.26 |
| **Concat** | 21.7 ± 0.17 | 28.1 ± 0.85 | 37.8 ± 3.09 | 26.6 ± 0.33 | **27.4** ± 0.29 | 33.4 ± 1.20 | 72.5 ± 0.30 | 35.4 ± 0.89 |
| **Data Selection** | 19.5 ± 0.76 | 25.7 ± 0.28 | 33.8 ± 2.23 | 25.9 ± 0.65 | 22.6 ± 1.24 | 30.6 ± 2.01 | 74.3 ± 0.28 | 33.2 ± 1.06 |
| **No-CL** | 20.1 ± 1.41 | 28.1 ± 0.85 | 37.8 ± 3.09 | 26.6 ± 0.33 | **27.4** ± 0.29 | 31.8 ± 0.28 | 72.2 ± 0.46 | 34.9 ± 0.96 |

Table 6: Balanced accuracy by loss.

# E  Index Importance Changes

|  | Index | Change Stages | Magnitude |
|---|---|---|---|
| **AN-Pairs** | Corrected TTR | Medium to late | 8.70% |
|  | Ratio of nx entity grid transitions | Medium to late | 7.89% |
|  | Semantic Richness | Medium to late | 7.86% |
| **GED** | Ratio of nx entity grid transitions | Medium to late | 3.18% |
|  | Lexical sophistication | Medium to late | 2.26% |
|  | Ratio of Coordinating Conjunction to Adjectives | Medium to late | 1.84% |
| **SNLI** | Ratio of Subordinating Conjunction to Adverbs | Early to late | 1.00% |
|  | Noun-subject transitions | Early to late | 0.79% |
|  | Number of topics (Weebit-based) | Early to late | 0.77% |
| **ANLI** | Verb sophistication | Medium to late | 0.98% |
|  | T-unit length | Medium to late | 0.96% |
|  | Log tokens over log sentences | Early to medium | 0.94% |
| **CoLA** | Object-noun transitions | Medium to late | 0.78% |
|  | TTR | Early to late | 0.75% |
|  | # Clauses | Medium to late | 0.75% |
| **RTE** | Adverb to adjective ratio | Early to late | 3.30% |
|  | # T-units | Early to late | 3.27% |
|  | Mean sentence length | Early to late | 3.15% |
| **SST-2** | Noun-subject transitions | Early to late | 1.76% |
|  | Coordinating conjunction per sentence | Early to late | 1.75% |
|  | Verbs per token | Medium to late | 1.72% |

Table 7: Top three moving linguistic indices.

Table 7 shows indices with maximum change in their $\rho$s between any two stages. Only the relative differences and ranking of $\rho$ values are important. Therefore, the table displays relative changes in the magnitude of the importance factors. The indices with a large change magnitude indicate that they are either influential at an early stage of training and drop in importance at the later stage, or vise versa. See our analysis on these indices in §3.7.