# OpenReview forum: "Ling-CL: Understanding NLP Models through Linguistic Curricula"
_EMNLP/2023/Conference — EMNLP 2023 Main_

### Official Review · Reviewer_jVRL · 2023-08-04

**Typos Grammar Style And Presentation Improvements:** I find section 3.4 remains a bit vagu…
**Soundness:** 4

**Excitement:**

4: Strong: This paper deepens the understanding of some phenomenon or lowers the barriers to an existing research direction.

**Paper Topic And Main Contributions:**

The paper investigates a new approach to curriculum learning, in which the importance of several linguistic indices are estimated and aggregated to provide estimates for the overall linguistic difficulty, in relation to the model's linguistic competence.
The authors find that their approach leads to improved performance compared to standard training (without curriculum) over several standard evaluation sets that require a high level of linguistic knowledge.

The paper also present an analysis of the key linguistic indices for grammar tasks that provide interesting insights into linguistic challenges and the factors relevant to model the respective tasks, finding that different factors take indeed different prominence for different tasks.

The main contributions are the incorporation of linguistic complexity information and the development of curricula based on linguistic
complexity, as well as the identification of the linguistic complexity indices that contribute most to different NLP tasks.



**Questions For The Authors:**

Languages: the paper only studies English, given that many resources are needed to estimate the linguistic indices. However, I wonder whether the results also hold for languages that are different from English.
Would it be possible to reduce the set of linguistic features to a subset that is available to more languages and then repeat the experiment at a smaller scale with more languages? (While this is certainly beyond the scope of this paper, a brief discussion about the applicability to other languages might be interesting...)

How do your results for the different evaluation sets compare to the results of related work?

**Reasons To Accept:**

The paper presents a novel approach to curriculum learned based on linguistic difficulty that goes beyond previous approaches with regard to the features used, and the way of modelling the curricula.

Furthermore, the paper provides an insightful analysis on the importance of different linguistic indices during different training stages.

Overall, the paper presents a very interesting idea, in combination with a meaningful evaluation.

**Reasons To Reject:**

There is not much information on the model and the experimental settings, only in the appendix.
I think it would be better to have some more information also in the main part of the paper.

A general downside of the approach is the dependency on linguistic knowledge generated from human experts, which will limit the amount of languages that can be modelled with this approach.

**Reproducibility:**

2: Would be hard pressed to reproduce the results. The contribution depends on data that are simply not available outside the author's institution or consortium; not enough details are provided.

**Reviewer Confidence:**

1: Not my area, or paper was hard for me to understand. My evaluation is just an educated guess.

---

> ### Author Rebuttal · Authors · 2023-08-28
>
> We thank the reviewer for their time, comments, and positive feedback on the merit of our work. In what follows, we try to address the reviewer’s questions and comments.
>
> **Q.  Would the results also hold for other languages beyond English? Could the experiment be repeated using a subset of linguistic features available to more languages?**
>
> ******A:****** In fact, some linguistic complexity indices are language independent (e.g., the commonly-used “word rarity” measure), which facilitates extending our approach to other languages, as suggested by the reviewer. We appreciate the reviewer’s insightful input, which formed the subject of our future research.
>
> **Q. How do your results for the different evaluation sets compare to the results of related work?**
>
> ******A:****** Our baseline models show good performance when linguistic complexity is not considered for evaluation, but their effectiveness drops when test samples are categorized according to linguistic complexity. Our approach improves the best-performing baseline performance by 4.5 absolute points in average F1 score, without compromising performance when linguistic complexity is not considered for evaluation.
>
> We follow the other comments provided by the reviewer on adding additional information on the model and the experimental settings (we will move some information from appendix to the the main part) and address all presentation suggestions.

---

### Official Review · Reviewer_5jzZ · 2023-08-05

**Paper Topic And Main Contributions:** 1. This paper proposes a series of ap…
**Typos Grammar Style And Presentation Improvements:** 1. Usage of hyphens and dashes should…
**Soundness:** 3

**Excitement:**

4: Strong: This paper deepens the understanding of some phenomenon or lowers the barriers to an existing research direction.

**Questions For The Authors:**

1. There are two variables, the number of samples and verb variation, so why can you say that "the accuracy trend indicates the linguistic index---verb variation---can describe the difficulty of ANLI samples to the model."



**Reasons To Accept:**

Several innovative methodologies are introduced in this work:
(1) The incorporation of correlation and optimization methods to gauge the importance of diverse linguistic indices.
(2) Utilization of either the maximum or weighted average for aggregation.
(3) Introduction of curricula encompassing time-varying sigmoid, moving negative-sigmoid, time-varying Gaussian function, etc.

The paper demonstrates adept composition and a well-structured layout. Readers can readily discern the research inquiries pursued and the corresponding actions taken by the authors.

The authors undertake a substantial array of experiments, bolstering the paper's foundation and substantiating its claims convincingly.

**Reasons To Reject:**

There are a series of methods proposed as mentioned before. However, it's worth noting that not all of these methods have been examined within the conducted experiments. For instance, the distinction between the correlation and optimization methods, as well as their respective impacts on system performance, is notably absent. This paper could potentially benefit from additional pages to accommodate a more extensive array of experiments, thereby offering a more comprehensive and in-depth understanding.

This concern extends to the treatment of linguistic indices. While multiple indices collaboratively contribute, the experimental focus narrows primarily to word rarity when evaluating different models. Yet, word rarity is not the top three most important linguistic indices in each dataset, which casts a certain degree of uncertainty upon the conclusions drawn.

Furthermore, certain expressions within the paper lack clarity. For instance, the term "competence-based" in line 350 and "indices filtering" in line 353 could benefit from a more explicit explanation. While the intended meanings can be inferred to some extent, providing a clear and concise elucidation would enhance the overall understanding of the content.

**Reproducibility:**

4: Could mostly reproduce the results, but there may be some variation because of sample variance or minor variations in their interpretation of the protocol or method.

**Reviewer Confidence:**

4: Quite sure. I tried to check the important points carefully. It's unlikely, though conceivable, that I missed something that should affect my ratings.

---

> ### Author Rebuttal · Authors · 2023-08-28
>
> We thank the reviewer for their time, comments, and positive feedback on the merit of our work. In what follows, we try to address the reviewer’s questions and comments.
>
> **Q. There are two variables, the number of samples and verb variation, so why can you say that "the accuracy trend indicates the linguistic index---verb variation---can describe the difficulty of ANLI samples to the model.”**
>
> ******A:****** We eliminate the number of samples as a variable by categorizing samples into evenly distributed groups based on their linguistic complexity and reporting the average performance of samples in each group. Then, we find the accuracy trend across these groups (determined by verb variation). Therefore, the accuracy trend is a direct consequence of verb variation as a linguistic complexity measure.
>
> **C. What is the distinction between the correlation and optimization methods and their respective impacts on system performance?**
>
> ******A:****** The main distinction between the two approaches lies in their scope: the correlation approach operates on the index level, whereas the optimization approach uses the entire set of indices. On average, across all datasets, the optimization approach outperforms correlation in our experiments. Also notably, on average, the argmax index aggregation outperforms the weighted average, and the filtered indices outperform the full list of indices. We will add the corresponding detailed results in the appendix.
>
> **C. The experimental focus narrows primarily to word rarity. Yet, word rarity is not the top three most important linguistic indices in each dataset**
>
> ******A:****** We use “word rarity” as a case study index to illustrate how a linguistic complexity index can facilitate identifying vulnerabilities in existing evaluation approaches for NLP tasks and how can such vulnerabilities be potentially addressed. In Figure 3 and line 509, we illustrate and discuss the relation between balanced accuracy and index importance, showing that it is not necessary for an index with low balanced accuracy to have a high \rho and vice versa.
>
> We address all presentation suggestions provided by the reviewer.

---

### Official Review · Reviewer_7FZF · 2023-08-05

**Soundness:** 3

**Excitement:**

4: Strong: This paper deepens the understanding of some phenomenon or lowers the barriers to an existing research direction.

**Paper Topic And Main Contributions:**

This paper proposes a method of characterizing data samples into linguistically informed complexity measures (such as verb complexity or word rarity) then weighting these data samples during model training to progressively emphasize data samples with increasingly higher complexity scores. This method is evaluated on seven NLP tasks including identifying entailment, inference making, grammar detection and others. The use of the linguistic complexity measures then offers an insight into what the NLP models use in terms of linguistic indices at different stages of training to perform the NLP tasks. The paper shows indices impact tasks differently, and we can learn from how the linguistic complexity of the data can impact model performance.

**Questions For The Authors:**

Question A: There are trends in Second Language Acquisition and Psycholinguistics research to cast doubt on the robustness of some linguistic complexity measures such as age of acquisition of forms, e.g., acquisition of morphology or syntactic complexity, e.g., surprisal in complex syntax isn't shared. Would removing the number of indices used in the method to more objective ones offer the same insights or change the outcome of the experiments?

**Reasons To Accept:**

- The paper nicely delineates how individual complexity measures contribute to performance on each task differently, and at what stage of the model training process; this may help others identify what impacts their model performance given what task they're working on.
- Well motivated approach of classifying data samples, then sequencing and weighting them during training - the insights how to use the data could be generalized to other methods
- Appendix provides helpful context of what indices were used and follow up experiments


**Reasons To Reject:**

- As the authors point out, linguistic complexity measures require expert knowledge to define and are not language agnostic. This method then may not generate over to other languages until similar measures of complexity are defined.
- It was not all together clear if the described experiments should be carried out by others in their own work such as identifying complexity of data to better understand their model performance, or if this is a more self contained experiment



**Reproducibility:**

4: Could mostly reproduce the results, but there may be some variation because of sample variance or minor variations in their interpretation of the protocol or method.

**Reviewer Confidence:**

3: Pretty sure, but there's a chance I missed something. Although I have a good feel for this area in general, I did not carefully check the paper's details, e.g., the math, experimental design, or novelty.

**Typos Grammar Style And Presentation Improvements:**

L282: Adjective-nous -> Adjective-noun (here, and elsewhere in the paper)
Table1: matthew's correlation -> uppercase

---

> ### Author Rebuttal · Authors · 2023-08-28
>
> We thank the reviewer for their time, comments, and positive feedback on the merit of our work. In what follows, we try to address the reviewer’s questions and comments.
>
> **Q. Would the exclusion of subjective linguistic indices in experiments, in favor of the objective ones, have a significant impact on the experimental outcomes?**
>
> ******A:****** While including more objective complexity signals is certainly preferable, we expect removing subjective indices to have a relatively minimal impact on our results. This is because our method is designed to automatically de-emphasize underperforming indices through the index importance factor \rho, which measures the correlation between validation loss (as a true measure of difficulty to model) and the index. So, indices that do not correlate well with the validation loss are assigned a weight of almost 0. It would be informative (and interesting) to determine which subjective indices are assigned low weights and which low-weight indices have the potential to be subjective indices.
>
> **C. Availability of complexity indices for extension to other languages beyond English.**
>
> ******A:****** We will update the limitations Section in the final version of this submission to highlight the reviewer’s point about extension to other languages.
>
> **C. Should the described experiments be carried out by others or if this is a more self-contained experiment?**
>
> ******A:****** We hope our framework and the corresponding tools to serve as a guide for assessing linguistic complexity for various NLP tasks and uncover the learning dynamics of the corresponding NLP models during training. While we conducted our analysis on seven tasks and extracted insights on the key indices for each task, NLP researchers have the flexibility to either build on our results or apply our approach to other NLP tasks to extract relevant insights. We will clarify this point in the final version of this paper.

---

### Meta-Review · Area_Chair_98eT · 2023-09-19

**Recommendation:** 4

**Metareview:**

This paper presents a multi-view curriculum learning framework, which include characterizing data samples using linguistic complexity measures and weighting these samples during model training based on their complexity scores. The paper also explores the impact of linguistic complexity measures on various NLP tasks and provides insights into how different linguistic indices affect model performance at different training stages. Reviewers have found the proposed approach well motivated, interesting and innovative; therefore, I recommend acceptance of this paper. Please also address the comments by reviewers on limitations.

---

### Decision · Program_Chairs · 2023-10-07

**Decision:**

Accept-Main

**Comment:**

This paper presents a multi-view curriculum learning framework, which include characterizing data samples using linguistic complexity measures and weighting these samples during model training based on their complexity scores. The paper also explores the impact of linguistic complexity measures on various NLP tasks and provides insights into how different linguistic indices affect model performance at different training stages. Reviewers have found the proposed approach well motivated, interesting and innovative; therefore, I recommend acceptance of this paper. Please also address the comments by reviewers on limitations.